# Recent Progress in the Detection of Surra, a Neglected Disease Caused by *Trypanosoma evansi* with a One Health Impact in Large Parts of the Tropic and Sub-Tropic World

**DOI:** 10.3390/microorganisms12010044

**Published:** 2023-12-26

**Authors:** Jeongmin Kim, Andrés Álvarez-Rodríguez, Zeng Li, Magdalena Radwanska, Stefan Magez

**Affiliations:** 1Laboratory for Biomedical Research, Department of Environmental Technology, Food Technology and Molecular Biotechnology KR01, Ghent University Global Campus, Incheon 21985, Republic of Korea; jeongmin.kim@ghent.ac.kr (J.K.); andres.alvarez.rodriguez@vub.be (A.Á.-R.); magdalena.radwanska@ghent.ac.kr (M.R.); 2Brussels Center for Immunology (BCIM), Department of Bioengineering Sciences (DBIT), Vrije Universiteit Brussel (VUB), 1050 Brussels, Belgium; zeng.li@vub.be; 3Department of Biochemistry and Microbiology WE10, Ghent University, 9000 Ghent, Belgium; 4Department of Biomedical Molecular Biology WE14, Ghent University, 9000 Ghent, Belgium

**Keywords:** Surra, *Trypanosoma evansi*, molecular diagnosis, point-of-care diagnosis

## Abstract

Surra is a wasting disease triggered by infection with *Trypanosoma evansi*, a protozoan blood parasite that causes mortality and morbidity in a broad spectrum of wild and domestic animals and occasionally humans. *Trypanosoma evansi* has the widest geographical spread among all pathogenic trypanosomes, inflicting significant worldwide economic problems due to its adverse effects on meat and milk production. For diagnosis, most endemic countries continue to rely on traditional parasitological and serological techniques, such as the analysis of blood smears by microscopy and the Card Agglutination Test for *T. evansi* (CATT/*T. evansi*). Although these techniques suffer from a limited positive predictive value (PPV), resource constraints in endemic countries often hinder the adoption of more advanced diagnostic tools such as PCR. This paper addresses diverse diagnostic approaches for identifying *T. evansi* and assesses their viability in field settings. Moreover, it underscores the urgency of transitioning towards molecular diagnostic techniques such as Loop-Mediated Isothermal Amplification (LAMP) and Recombinase Polymerase Amplification (RPA) for dependable high-PPV point-of-care (POC) diagnostics. Finally, this review delves into strategies to enhance and refine next-generation diagnostics for Surra as part of a One Health approach.

## 1. Introduction

*Trypanosoma evansi* is a flagellated protozoan blood parasite that can naturally infect a wide range of animal hosts, giving rise to a wasting disease called Surra. This type of animal trypanosomosis is found in tropical and subtropical regions around the world and affects both domestic and wild animals [1,2]. The name Surra is often used in Africa and Asia, meaning “bellybutton” in Arabic, to describe the frail and emaciated conditions of the infected animals [3,4]. Other names such as “Mal de Caderas” or “Derrengadera,” which mean “bad hip” and “exhausted,” are more commonly used in South America, including Argentina, Brazil, Paraguay, and Venezuela [5].

*Trypanosoma evansi* has the widest host range among all trypanosome species, and its transmission is facilitated by numerous genera of biting flies, including *Tabanus*, *Stomoxys*, and *Haematopoda* [6,7]. Camels, cattle, buffalo, goats, equines, sheep, and dogs can all serve as parasite reservoirs, with severe infections observed mostly in camels, equids, cattle, and buffaloes. The distribution of *T. evansi* is not only species-dependent but also geography-dependent. For example, in Africa and the Middle East, camels are the major hosts, while in Asia, water buffaloes and elephant infections are most common, especially in the Philippines and India [8]. In South America, the vampire bat species *Desmodus rotundus* is a prominent vector and host, while deer, wild pigs, rodents, and other animals can be involved in the transmission of rare infections in Australia and Europe [1,3,9]. Such host diversity provides significant potential for *T. evansi* to thrive and spread globally.

Surra is severely neglected in many parts of the world despite the damage it brings to the economy, killing thousands of animals annually and occasionally even causing a fatal form of atypical human trypanosomiasis (aHT) [4]. Areas largely affected by Surra are Asia, South America, Africa, and the Middle East. However, the remarkable transmissibility of *T. evansi*, which is facilitated by numerous insect vectors, has historically resulted in outbreaks of *T. evansi* in countries in Europe, including Spain and France, caused by the unintended importation of infected camels from the Canary Islands [4,7]. Hence, it is clear that *T. evansi* poses both a direct and indirect threat to human health, with the latter related to the danger to economically important livestock herds. Therefore, the detection or diagnosis of Surra and *T. evansi* aHT should both be considered crucial components of a One Health approach in the developing world.

The paper addresses significant challenges in the treatment and control of Surra. This includes the reliance on trypanocides in the absence of a developed vaccine, as well as hurdles in pathogen and vector control. It explores the often-overlooked prevalence of Surra, underscoring the critical need for accurate diagnosis to enable early detection for optimal treatment, thereby minimizing the risk of drug resistance and toxicity. Furthermore, it explores recent progress in various diagnostic methods for *T. evansi*, including Loop-Mediated Isothermal Amplification (LAMP) and Recombinase Polymerase Amplification (RPA). A crucial point is made regarding the necessity for point-of-care diagnostic tests, especially in resource-limited settings.

## 2. Transmission and Pathology

*Trypanosoma evansi* belongs to a group of salivarian trypanosomes, which include other subspecies such as *T. brucei brucei*, *T. brucei gambiense*, *T. brucei rhodesiense* (the latter both causing human sleeping sickness), and *T. equiperdum* (causing Dourine), as well as *T. congolense* and *T. vivax* (both causing Nagana) [10]. Due to its close evolutionary proximity to *T. b. brucei*, the parasite is also referred to as *Trypanosome brucei evansi*. Salivarian trypanosomes differ from stercorarian trypanosomes such as *T. cruzi*, *T. rangeli, T. lewisi*, and *T. theileri*, which are transmitted via the feces of the insect vector [11].

Known to have evolutionarily diverged from *Trypanosoma brucei brucei* in Africa, *T. evansi* adapted to mechanical transmission, setting itself apart from other species that rely on cyclical transmission by tsetse flies [9,12]. By losing its maxicircle kinetoplast DNA, *T. evansi* was no longer limited to the tsetse belt in Sub-Saharan Africa and gained a pass to spread to other continents [13,14,15]. Transmission mainly occurs by numerous genera of biting flies, including *Tabanus*, *Stomoxys*, *Haematopoda*, *Chrypsos*, and *Lyperosia* [6]. Tabanids, in particular, play a significant role in the transmission of Surra as a determined feeder with well-adapted mouthparts for both sucking and lapping blood. Their mouthparts are optimal to facilitate the transmission of *T. evansi* parasites by trapping a blood film and keeping it from drying. A feeding time of 5 s is enough for transmitting parasites to the mammalian host, and the probability of transmission is higher for a shorter interval between infective feeds [6]. The transmission of *T. evansi* is mainly carried out mechanically, as illustrated in Figure 1. Besides mechanical transmission, *T. evansi* can also be transmitted directly through copulation, the consumption of milk, or by ingesting freshly killed infected animals in the case of a carnivorous host [1]. Infection with *T. evansi* can be acute, with high mortality in some cases, and chronic in others. It is characterized by fever, emaciation, stiffness of the limbs, nasal and ocular bleeding, anemia, abortion, and death within weeks or months when left without treatment for both wild and domestic mammals [3,16,17]. A loss of condition and reproductive performance can be observed in chronic forms of the disease, which can also lead to neuropathy. Immunosuppression is another key symptom that puts animals in danger of multiple opportunistic diseases, which pose a detrimental threat to meat and milk production [18].

Although less common, infections in humans have been observed in countries such as Vietnam, India, and Sri Lanka. Some human trypanosomiasis cases were observed in which the infected individual showed symptoms of intermittent cyclical fever, discomfort, and drowsiness for over five months [19,20]. To date, the cause of these infections remains unknown. In 2004, a 45-year-old herdsman was diagnosed with a *T. evansi* infection in Chandrapur, India. The cause of the infection was found to be a frameshift mutation in both alleles of APOL1, a lipoprotein located in the plasma that kills trypanosomes by lysosome swelling [21]. However, such a mutation was not found in the case of the 38-year-old woman, who in 2015 was diagnosed with a *T. evansi* infection in Dong Nai Province in southern Vietnam [21]. In this case, the only identified risk factor was that the patient had given birth 2–3 months earlier. Frequent human contact with *T. evansi*-infected animals may elevate the risk of its zoonotic potential, contributing to the emergence of non-African or so-called atypical human trypanosomiasis [22]. Therefore, controlling animal parasitemia is essential to prevent its further spread to humans across different continents. 

## 3. Control and Treatment of Surra

The control of Surra can be largely divided into pathogen control and vector control. Curative and preventive trypanocides are heavily relied upon for pathogen control since a vaccine against *T. evansi* has not yet been developed due to their meticulous evasion of the host immune system through multi-line defense strategies. Their genome contains over 1000 genes that encode for variable surface glycoproteins (VSGs), which can be turned on and off to give the parasite a new VSG coat to circumvent the host immune system, often causing relapses of parasitemia [1,3,23]. The host immunoglobulin response also becomes helpless since trypanosomes can inhibit complement-mediated lysis by removing bound immunoglobulins from their surface [24,25,26,27,28]. On top of that, the disruption of the host B cell compartment by *T. evansi* further increases the difficulty in combatting these parasites [29,30,31,32,33].

The prevalence of Surra is often overlooked due to limitations in the correct and efficient diagnosis of the disease. The clinical signs and parasitemia of Surra vary, and hosts often present mild, chronic forms of the disease. However, even when present in low numbers, it is sufficient to suppress the host immune response, rendering the latter vulnerable to other diseases [23]. Furthermore, when *T. evansi* starts to hide in the host nervous system in the latent stage of the infection, it makes detection even more challenging. Since, at this stage, parasites are hardly present in the peripheral blood, more invasive approaches are needed for diagnosis, such as cerebrospinal fluid (CSF) collection [34]. 

For the treatment of blood-stage Surra, suramin is most commonly used. This drug is administered intravenously and works by interfering with the metabolism and energy production of the parasite. Other treatments include melarsomine dihydrochloride (Cymelarsan^®^) and quinapyramine. In Asian countries, diminazene aceturate (Berenil^®^) is often used to treat Surra during the blood stage. However, drug resistance often emerges with these parasites due to under-dosed treatments. Moreover, the toxic side effects of these drug treatments can be highly detrimental to the animal host [22]. Therefore, early and accurate diagnosis is key for better treatment and to prevent false-positive results that could force healthy animals to undergo toxic treatment. This highlights the need for simple but reliable point-of-care (POC) diagnostic tests, especially for countries that lack resources and sterile lab settings for complex molecular diagnostic methods. 

When discussing the efficacy of diagnostic methods, specificity and sensitivity are commonly emphasized, despite their limitations when implemented in real-world settings (see Equation (1)). In cases of low prevalence, sensitivity becomes heavily influenced by the number of false-negative cases, which compromises its primary purpose of measuring the test’s accuracy in producing positive results. Likewise, in scenarios with high disease prevalence, false-positive cases can distort the specificity of a diagnostic method, rendering it an unsuitable measure for the test’s validity in producing negative results. To address this issue, positive predictive values (PPVs) and negative predictive values (NPVs), as shown in Equation (2), should be utilized as more practical measures to determine an effective intervention method based on a test result. These values adjust to varying prevalence conditions, unlike sensitivity and specificity values [34]. Hence, greater emphasis should be placed on PPVs, as they indicate the likelihood of a positive score in a diagnostic test identifying a truly positive patient, which is particularly important in low-prevalence scenarios. Unfortunately, PPVs are currently hardly mentioned in the *T. evansi* diagnostic literature.
(1)$$Sensitivity=\frac{\sum True(+)}{\sum True\left(+\right)+\sum False(-)}\;\;\;Specificity=\frac{\sum True\left(-\right)}{\sum True\left(-\right)+\sum False\left(+\right)}$$
(2)$$PPV=\frac{\sum True\left(+\right)}{\sum True\left(+\right)+\sum False\left(+\right)}\;\;\;\;\;NPV=\frac{\sum True\left(-\right)}{\sum True\left(-\right)+\sum False\left(-\right)}$$

## 4. Parasitology- and Serology-Based Diagnostic Methods

### 4.1. Microscopy-Based Detection Methods

Microscopy can provide insights into the diagnosis of Surra by identifying *T. evansi* parasites in blood samples. Blood smears are widely used for microscopic examination, where a drop of collected blood is placed on a microscope slide. The smear is air-dried, fixed, and commonly stained with Giemsa stain. The pinkish–purple stain helps highlight the appearance of parasites under a microscope [2]. It is a simple and inexpensive method to diagnose Surra and, hence, is widely used in the field [35]. Blood smears can detect both Type A and Type B *T. evansi*, which differ, respectively, by the presence and absence of the minicircle or Rode *Trypanozoon* antigenic type 1.2 (RoTat 1.2) VSG gene expression [22]. Type A is most common amongst *T. evansi* isolates, while Type B is only found in limited areas of Africa, including locations such as Kenya, Egypt, and Ethiopia [36,37]. Parasitemia with *T. evansi* can be high, reaching up to 1 × 10^6^ parasites per 1 mL of blood. This is higher than in other species, such as *T. brucei gambiense*, where parasitemia levels during a chronic infection typically do not exceed 1000 parasites per 1 mL of blood [20].

While high parasitemia might be considered advantageous for blood smear detection, parasitemia with *T. evansi* fluctuates intermittently, frequently falling below the limit of detection (LOD) of 10,000 parasites per 1 mL of blood [34]. Hence, reliable detection is only guaranteed during the acute phase of the disease, with a low sensitivity ranging from 25% to 50%, resulting in false-negative results 50% of the time [5,34,35,38]. Other limitations also exist, including the inability to diagnose latent or chronic infections, an ambiguous cut-off for observation time, and the failure to distinguish one species of *Trypanosoma* from another. Particularly, in practice, chemotherapeutic treatment using trypanocides is usually the only option available for controlling the disease. However, resistance to all drugs has been reported, with different parasite species having different levels of sensitivity to the available medications [39]. Given the worldwide distribution of *T. evansi*, this parasite is present in regions where other trypanosome species are also prevalent. Therefore, the accurate identification of the trypanosome species is crucial for making optimal treatment decisions. Furthermore, microscopy sample processing has a relatively lengthy execution time, despite the simplicity of the method, and is not always suitable for field settings as it requires microscopy equipment. The blood smear method is also limited to specifically trained personnel and cannot be performed by the general public [40]. 

Overall, parasitological methods, such as microscopy and hematocrit centrifugation techniques (HCT), face a significant limitation as they cannot confirm the identity of the parasite at the species level and often exhibit low sensitivity.

### 4.2. Serology-Based Detection Methods

Serological methods detect antibodies and proteins produced by the immune system in response to pathogens. RoTat 1.2-based diagnostic tests detect antibodies in the serum that indicate a Type A *T. evansi* infection. These tests include the Card Agglutination Test for *T. evansi* (CATT/*T. evansi*), the Latex Agglutination Test for *T. evansi* (LATEX/*T. evansi*), and the Enzyme-Linked Immunosorbent Assay for *T. evansi* (ELISA/*T. evansi)* (see Table 1) [41]. The variant antigen type (VAT) of a trypanosome is determined by the VSG, which is highly immunogenic and elicits an antibody response in the host for opsonization, agglutination, and trypanolytic activity.

CATT/*T. evansi* is a rapid, direct agglutination test that detects the presence of *T. evansi*-specific antibodies in the host blood. This test involves placing a small amount of host serum on a card along with Coomassie-stained and freeze-dried *T. evansi* RoTat 1.2 trypanosomes. If antibodies against *T. evansi* are present in the serum, they will bind to the VSG antigens on the surface of the parasites, causing agglutination. This is used as a visual indicator for the presence of specific antibodies due to a *T. evansi* infection. The test is fast, easy, and well-suited for field applications, which significantly improves the blood smear test. Nonetheless, CATT/*T. evansi* suffers to an extent from poor specificity and cross-reactivity with antibodies against closely related parasites such as *T. cruzi* [42,43]. To prevent antibody/antigen degradation in blood samples, careful preservation is required by refrigerating or freezing the samples appropriately. One of the biggest limitations of this test is that it cannot distinguish between previous and current infections. This means that previous infections, repeated exposures (without disease development), or unrelated polyclonal B cell activation (triggered by another infectious agent) could also lead to the release of host antibodies, yielding a positive result [22,42]. Other serological methods, such as LATEX/*T. evansi* (rapid indirect agglutination method) and ELISA/*T. evansi*, exhibit similar problems as CATT/*T. evansi* due to the use of host antibodies as a signal for infection. In addition, while these methods can detect infections during the chronic stage, they are not optimized for early detection during the prepatent stage of the infection when only a small amount of target analytes are present. [5].

False-positive diagnoses could lead to detrimental consequences, as demonstrated by Njiru et al., who reported an example in which 252 animals would have undergone drug treatment instead of 144 animals if CATT/*T. evansi* were used for diagnosis instead of PCR [16].

**Table 1 microorganisms-12-00044-t001:** List of serological tests for the diagnosis of *T. evansi*.

Species	Name of the Assay	Target	Reference
*T. evansi*	Double immunodiffusion (DID) test	Whole cell soluble *T. evansi* antigen	[44]
*T. evansi Type A*	Card agglutination test for trypanosomosis/*T. evansi* Rotat1.2 (CATT/*T. evansi*)	VAT *T. evansi* Rotat1.2	[44]
*Trypansomal*	IgG-ELISA	Trypanosomal Abs	[45]
*T. evansi*	IgG-ELISA	2G6 Ag-ELISA (70 kDa antigen) Tr7 Ag-ELISA (15 kDa antigen)	[46]
*Trypanozoon* spp. *T. vivax*	Ag-ELISA	TeGM6-4r	[47]
*T. evansi*	TeCA-ELISA	*T. evansi* crude antigens	[47]
*T. evansi*	Suratex	Trypanosome-circulating Ags	[48]
*T. evansi*	IFAT	Anto-*T. evansi* Abs	[49]
*T. evansi*	LATEX/*T. evansi*	Rotat 1.2 VSG	[50]
*T. evansi*	ELISA	Rotat 1.2 VSG	[50]
*T. evansi*	ELISA	Excretory-secretory Ags	[51]
*T. evansi*	IgG-ELISA	Anti-*T. evansi* Abs	[52,53]
*T. evansi*	Ab-ELISA/rISG75	Anti-*T. evansi* Abs	[54]
*T. evansi*	Ag-ELISA	TEA 1/23.4.6	[55]
*T. evansi*	Immune trypanolysis test (TL)	VAT RoTat 1.2	[56,57]
*T. evansi type A*	Surra Sero K-SeT	VAT RoTat 1.2	[57]
*T. evansi*	LFA	Anti-*T. evansi* Abs	[58,59]
*T. evansi*	IgM-ELISA	Anti-*T. evansi* Abs	[60]
*T. evansi* *T. brucei*	Ag-ELISA	Circulating Ags	[61]

## 5. Molecular Diagnostic Methods

### 5.1. Polymerase Chain Reaction

Invented in the 1980s, the Polymerase Chain Reaction (PCR) revolutionized molecular detection methods using nucleic acids (Figure 2) [62]. To perform amplification, blood samples are obtained from the field and well-preserved for DNA extraction. The visualization of amplified products is most commonly carried out by agarose gel electrophoresis [63].

Numerous genes are employed to diagnose trypanosomes, including *Trypanosoma brucei* repeat (TBR1/2) and ESAG6/7. These multicopy genes are present in the *Trypanozoon* subspecies, which comprises *T. evansi*, *T. brucei*, and *T. equiperdum*. TBR plays an essential role in antigenic variation, while ESAG encodes the transferrin receptor complex [35]. Generally, TBR1/2 primers exhibit higher sensitivity than ESAG6/7 primers [64]. Ribosomal RNA (rRNA) genes are also frequently used, as their high conservation allows for the distinction of closely related trypanosome species [42]. Internal transcribed spacer (ITS) regions belong to trypanosome rRNA and are flanked by highly conserved segments, which can be exploited for primer design. PCR primers can also be designed so that each species generates a unique length of PCR amplicon. ITS-1, for example, exhibits variability even among closely related species and is often used in diagnosing *Trypanozoon* subspecies [43]. To distinguish *T. evansi* from other members of the subgenus *Trypanozoon*, primers derived from the RoTat 1.2 VSG sequence are commonly used, as they are specific to Type A *T. evansi* [17]. Other *T. evansi* types do not produce amplicons with this primer set, including Type B *T. evansi*. For detecting Type B *T. evansi*, the EVAB and VSG JN 2118Hu genes can be used as targets (see Table 2) [43,65]. In terms of specificity, ITS-1 shows a specificity higher than 99% for *Trypanozoon* subspecies, and RoTat 1.2 exhibits a specificity of 86% for Type A *T. evansi* [35,43].

PCR can be coupled with fluorescent dyes to offer more in-depth information about an infection. For example, real-time PCR using SYBR green can provide information about the parasitemia level, whereas regular PCR only indicates the infection status [2]. However, given the rapid variability in parasitemia levels, real-time PCR results may not necessarily offer a clinically determinative number. Additionally, a high-resolution melting analysis can be coupled with PCR (HRM-qPCR) to successfully differentiate one species of *Trypanosoma* from another. This method takes advantage of DNA sequences having unique melting temperatures. A melting curve can be obtained using a saturated dye, such as SYTO-9, offering useful information about the unique parasite identity based on its sequence length or GC content [66]. Mixed infections can also be characterized by employing this method.

One of the greatest advantages of PCR is its high PPV, specificity, and sensitivity, with the limit of detection ranging from 1 to 10,000 parasites per 1 mL of blood [34]. This ensures a reliable diagnosis of Surra in both the prepatent and chronic phases of the disease. Its sensitivity is superior to that of parasitological tests, such as blood smears, especially during the chronic stages of infection [35,63]. Another significant advantage of PCR is its ability to detect active infections because the target DNA does not persist in the host’s body for an extended period, meaning that any detected DNA signifies a recent infection.

However, PCR also has disadvantages, such as its requirement for a lab setting. It is also time-consuming, with a reaction time of 1 to 2 h, and demands expensive reagents and well-trained staff to perform the test [34]. Therefore, despite the heightened PPV/sensitivity of PCR in detecting *T. evansi* compared to that of CATT/*T. evansi* and other serological tests, it has substantial limitations for field applications and cannot be widely used in endemic countries with limited resources [16]. To overcome such limitations, isothermal amplification methods have become available, allowing for the simpler application of diagnosis tests.

### 5.2. Isothermal Amplification Methods: Loop-Mediated Isothermal Amplification

#### 5.2.1. Loop-Mediated Isothermal Amplification Assay

Loop-Mediated Isothermal Amplification (LAMP) is an isothermal amplification method invented in the year 2000 that can amplify DNA molecules to 10^9^ copies within a short processing time [67,68]. This technique requires two primer sets, with an additional loop primer set often being used to increase the specificity, efficiency, and speed of the amplification process [69]. This FDA-approved method is conceptually similar to PCR and requires blood samples to be obtained and well preserved for accurate results [70]. The reaction mix contains a DNA polymerase, dNTPs, and specifically designed primer sets. As the primers bind to the target DNA, a characteristic dumbbell-shaped loop structure is created, enabling self-priming and an exponential amplification cycle (Figure 2). A cauliflower-like structure bearing multiple loops of DNA is generated post-amplification [68,69]. *Bst* DNA polymerase is preferred over *Taq* polymerase since its performance is less hindered by impurities such as hemoglobin and myoglobin that may be present in the DNA sample [71]. *Bst* DNA polymerase also exhibits robust strand displacement activity, which is essential for the continuous amplification of target DNA without a denaturation step.

In an LAMP assay, DNA amplification occurs isothermally at temperatures ranging from 58 to 65 °C [72]. Amplified products can be monitored in real-time using fluorescent dyes and turbidity measurements or visualized with gel electrophoresis. Using fluorescent dyes, the level of parasitemia can also be determined by comparing the fluorescence signal to a standard curve generated from various known concentrations of parasites. This is based on the principle that the amount of fluorescence produced is directly proportional to the initial amount of target DNA present in the sample. Similarly to PCR, conserved DNA sequences in trypanosomes, including ITS-1/2, 18S rRNA, RoTat 1.2 VSG, and RIME, can be used to detect *T. brucei*, *T. congolense*, and *T. evansi* (see Table 2) [49,53,55,73,74].

The sensitivity of a LAMP reaction can be improved by selecting regions of parasite DNA with a high copy number when designing primer sets [75]. This is because a high copy number allows for higher numbers of DNA copies to be generated in each cycle, enabling parasite detection even with a low parasite count. On the other hand, it is important to ensure that the target sequence is specific to the parasite to be detected. If not, the validity of the detection method is undermined. LAMP sensitivity can be further improved by using detergent during the sample preparation step. This helps release DNA from pathogen cells before the analysis, improving the sensitivity by 100- to 1000fold [76,77].

The advantages of a LAMP assay include not requiring a thermal cycler, which makes it far less expensive than performing PCR. It also exhibits a high specificity and sensitivity of 95% and 90%, respectively. Its LOD lies between 1 and 1000 parasites per mL of blood, similar to that of PCR [13,34]. Hence, LAMP can be more sensitive than microscopy or PCR-based detection methods [22]. Its running time ranges between 30 and 60 min, which is much shorter than PCR, and detection can be carried out visually by measuring the turbidity or fluorescence in a reaction mixture [67]. Moreover, LAMP can be conducted in a dry format, enabling detection with the naked eye through the incorporation of dyes such as SYBR Green I [78,79].

There are also some disadvantages to the LAMP procedure, such as the need for a complex primer design and a high susceptibility to contamination. Theoretically, high specificity is ensured in a LAMP assay since the amplification is achieved using 2–3 primer sets to recognize the target sequence [80]. However, careful design is necessary to facilitate a precise interaction between the 2–3 primer sets, which may require intensive optimization and potentially result in added expenses [81]. Additionally, the slight contamination of a negative sample with a positive one could yield a false-positive LAMP result. A high DNA concentration above 200 ng could also inhibit the amplification of the target DNA [80]. There is also a high chance of aerosol contamination in open-format LAMP assays [22]. The running time of LAMP is significantly shorter than that of PCR but is still too long for practical field applications. Despite numerous advantages, a laboratory condition is still required to ensure sterile test performance, presenting numerous hurdles for it to be a suitable POC test.

#### 5.2.2. Recombinase Polymerase Amplification

Recombinase Polymerase Amplification (RPA) is a novel isothermal amplification method capable of exponentially amplifying as few as 1–10 DNA copies [70]. It was invented in 2006, showing great potential for replacing PCR for simpler and faster operations [82]. This method does not require the thermal denaturation of DNA and has been used to detect bacteria, parasites, and viruses [83]. Multiple zoonotic diseases, such as tuberculosis, rabies, avian influenza, and SARS, can be diagnosed using RPA methods [84].

The main components of an RPA include a set of primers, recombinase proteins, single-stranded binding (SSB) proteins, and strand-displacing DNA polymerase. The initiation of an RPA assay requires a primer set to form a complex with the recombinase enzymes. The recombinase and SSB proteins aid in the unwinding of the double-stranded DNA and facilitate the binding of the primers to the target DNA sequence through hybridization (Figure 2). Similar to PCR and LAMP, the target DNA sequence that is most often used for the detection of *T. evansi* is RoTat 1.2 VSG (see Table 2) [40]. Once the primers are bound, a strand-displacing DNA polymerase begins to synthesize new DNA strands complementary to the target sequence, and a cyclical repetition of this process leads to exponential amplification. Both DNA and RNA can be amplified using this procedure, and the resulting products can be monitored at the endpoint as well as in real-time using appropriate probes (Figure 3) [83]. Endpoint visualization is achieved through gel electrophoresis or lateral flow assays.

**Table 2 microorganisms-12-00044-t002:** Molecular diagnostic methods, primers, and target genes used for the detection and amplification of *T. evansi*.

Species	Name of the Assay	Primer Name	Oligonucleotide (5′-3′)	Gene Target	Reference
*Trypanosoma* spp.	PCR	CF BR	CCGGAAGTTCACCGATATTG TGCTGCGTTCTTCAACGAA	ITS1	[85]
*Trypanozoon* spp.	PCR	F R	ACATTCCAGCAGGAGTTGGAG CACGTGAATCCTCAATTTTGT	ESAG 6/7	[86]
*Trypanozoon* spp.	PCR	21mer 22mer	TGCAGACGACCTGACGCTACT CTCCTAGAAGCTTCGGTGTCCT	Repetitive sequence probe pMUTec 6.258	[87]
*Trypanozoon* spp.	PCR	TBR1 TBR2	GAATATTAAACAATGCGCAG CCATTTATTAGCTTTGTTGC	Minisatellite DNA	[88]
*T. evansi* and *T. brucei*	PCR	TBS-01 TBS-02	CGAATGAATAATAAACAATGCGCAGT AGAAGGATTTATTAGCTTTGTTGC	Conserved regions of *T. brucei* and *T. evansi* genome	[89]
*T. evansi*	PCR	TR3 TR4	GCGCGGATTCTTTGCAGACGA TGCAGACACTGGAATGTTACT	Repetitive nucleotide sequences of *T. evansi*	[90]
*T. evansi*	PCR	TeD-ISGF TeD-ISGR	CAGCCGGTGAGTGAAGAAA CTACGGCCCCTAATAATAAAGAAC	ISG-75	[91]
*T. evansi*	PCR	NRP1 NRP2	CGAATGAATATTAAACAATGCGCAGT AGAACCATTTATTAGCTTTGTTGC	Nuclid Repeat	[92]
*T. evansi*	PCR	MP1 MP2	CAACGACAAAGAGTCAGT ACGTGTTTTGTGTATGGT	Minicircle DNA	[92]
*T. evansi*	PCR	EVA1 EVA2	ACATATCAACAACGACAAAG CCCTAGTATCTCCAATGAAT	Minicircle DNA	[93]
*T. evansi*	PCR	DITRYF DITRYR	CGACCAGCCAGAACGAGCAGAAT CTTGTCGATCGAGTTGACGGT	VSG	[94]
*T. evansi* (Type A)	PCR	F R	GCGGGGTGTTTAAAGCAATA ATTAGTGCTGCGTGTGTTCG	Rotat 1.2 VSG	[17]
*T. evansi* (Type B)	PCR	F R	TTCTACCAACTGACGGAGCG TAGCTCCGGATGCATCGGT	VSG JN 2118Hu	[65]
*T. evansi* (Real-time PCR)	PCR	TeRoTat920F TeRoTat1070R	CTGAAGAGGTTGGAAATGGAGAAG GTTTCGGTGGTTCTGTTGTTGTTA	RoTat 1.2 VSG	[95]
*T. evansi* (Real-time PCR)	PCR	TeRTF TeRTR	GGAAGCAAAAGTCGTAACAAGG CCCATGTCAAACGGCATATAG	ITS1	[2]
*T. evansi* (TaqMan PCR)	PCR	F R Probe	ATAAATTGCACAGTATGCAACCAAA CATCCCTCATCTCCCATGTCA 6FAM-ACGGCATATAGAAACACA-MGBNFQ	ITS1 internal	[96]
*T. evansi* (Type A)	LAMP	F R F3 B3 FIP (F1c + F2) BIP (B1c + B2) LF LB	CAAAACTAACAGCCGTTGCAGCG AGTTCCGGTACCTTCTCCATTTC GTAGGAAGCAACACCTGCG TTGATTAGTGCTGCGTGTGT TGCGAGGTGCACCTTGATGTTGAAGCAATAACCGGCAACGAC GAAGGCAAAGTTGACGACCAGCTGTGGTGTGCTTTTCCTTGT GCGATTTTGATCCCGCCG CAGAACGAGCAGAATTTTCCA	Rotat 1.2 VSG	[22]
*Trypanosoma* spp.	LAMP	F3 B3 FIP (F1c + F2) BIP (B1c + B2) LF LB	CTGTCCGGTGATGTGGAAC CGTGCCTTCGTGAGAGTTTC GGAATACAGCAGATGGGGCGAGGCCAATTGGCATCTTTGGGA AAGGGAGACTCTGCCACAGTCGTCAGCCATCACCGTAGAGC GCCTCCCACCCTGGACTC AGACCGATAGCATCTCAG	RIME	[69]
*T. evansi* (Type A)	LAMP	F3 B3 FIP (F1c + F2) BIP (B1c + B2)	TCACAACAAGACTCGCACG GGGCTTTGATCTGCTCCTC TCAGAAGCGTCGAGCTGGGATTTTATCGACAATGCCATCGCC CGCAAGTTCCTGTGGCTGCATTTTTTCCCAAGAAGAGCCGTCT	PFR A1	[67]
*T. evansi* (Type B)	LAMP	F3 B3 FIP (F1c + F2) BIP (B1c + B2) LF LB	CCAATCAAAGACGAGCGG TGGTTTGTGAGGCCGCAG CGGATGCATCGGTGATGCAATCACTACTGCATCAAGGGAAGC ATCCAGCACCTCGGAACAGCTCTCGGCAACCAGATCGG GTTCACGTGCCTCCGCTTC ACGTAGCGGGAAAATACGC	VSG JN 2118Hu	[80]
*T. evansi* (Type A)	RPA	TevRPA-Fw TevRPA-Rv	CACCGAAGCAAGCGCAGCAAGAGGGTTAGCA GTAGCTGTCTCCTGGGGCCGAGGTGTCATAG	Rotat 1.2 VSG	[40]

There are two types of RPA assays: exo-RPA and lateral flow (LF) strip RPA. Exo-RPA is a real-time RPA assay that utilizes fluorescent probes to rapidly detect target genes. The fluorescent probe is placed near a quencher so that fluorescence is only released after the quencher is cleaved from the fluorophore following the detection of target DNA. The fluorescence amplification curve is used to gather quantitative information for further analyses [83]. Exo-RPA has advantages in terms of speed compared to the LF strip and a lower risk of aerosol contamination, as the reaction tubes do not need to be kept open. The only downside is that an instrument is required to detect the fluorescence [84].

LF strip RPA combines the original RPA technology with chromatography test strips and immunoassays to detect amplified DNA visually. The operating mechanism is similar to that of exo-RPA as it requires an *Nfo* endonuclease to release the blocker from a 5′ FAM-Spacer-Blocker 3′ probe following the successful binding of the 5′ FAM-Spacer fragment to the target DNA sequence. As the sequence containing FAM and biotin on opposite sides is obtained, the amplified products are able to bind to the anti-biotin antibodies on the test line. For a negative test result, only one band will appear on the control line, while both the control and test lines will show a band in the case of a positive result. This technology was applied in 2020 to develop an LF-RPA test that allows for the diagnosis of Type A *T. evansi* in a simple field setting with a rapid execution time of only 15 min. The LF strip has a user-friendly readout and has also demonstrated high sensitivity, allowing successful diagnosis with just 100 fg of *T. evansi* DNA [40]. The rapid readout of results without the use of an instrument is a significant strength of the RPA-LF strip test. Nonetheless, the test might suffer from reduced stability as well as a high risk of aerosol contamination. The latter renders the method prone to false-positive results, negatively affecting the PPV [84].

A characteristic advantage of RPA tests, regardless of their type, is their rapid execution time, ranging from 10 to 20 min, which is much quicker than that of both LAMP- and PCR-based methods. They are cost-effective and require minimal equipment for operation, as the reaction is also performed isothermally at an ambient temperature range between 37 and 42 °C [70]. The fact that the operating temperature does not deviate significantly from the natural body temperature of most mammals also makes this method suitable for field applications. The LOD for this method is 100 parasites per 1 mL of blood, which is 10fold more sensitive compared to PCR and LAMP. Another advantage of RPA is that it can be directly performed on serum samples, even in the presence of hemoglobin, ethanol, and heparin, which act as inhibitors in PCR [70]. Despite the risk of aerosol contamination, the LF strip test is user-friendly and can also be multiplexed to detect other *Trypanosoma* species. These qualities make RPA methods suitable as a POC test.

Unfortunately, the RPA method also has weaknesses in that it lacks special software for primer design and that there are fewer reagent kits available [83]. Additionally, in the case of the *T. evansi* RPA, the test is prone to non-specific amplification, meaning that DNA molecules with high similarity to the target DNA sequence will also often be amplified [34]. To address this issue, CRISPR-Cas technology can be coupled with RPA to exploit the specific binding of the CRISPR-Cas system to the target sequence, thereby reducing background noise or non-specific amplification that can occur in RPA reactions [34]. CRISPR-Cas combined with RPA allows all-in-one tube reactions to be performed at around 37 ºC and even reach lower LOD values than RT-RPA or LF-RPA. However, it also requires more time for detection, and the overall costs can be higher due to the use of ssDNA/RNA reporters [84].

Finally, there is also an electrochemical RPA (eRPA), which combines RPA with the detection of an electrochemical signal that correlates with the amount of target DNA present in the sample, allowing for real-time quantification. Its advantages are decreased analysis time and a reduced chance of contamination, but it also suffers from low amplification efficiency [84].

## 6. Future Perspectives

As the global climate rapidly changes, favoring the growth and redistribution of insect vectors, the need for effective diagnostic tools to enhance the control of vector-borne diseases becomes increasingly urgent. Hence, the investigation of additional methods to curb the spread of Surra is imperative. Such methods can be as simple as separating the breeding locations of bovines and equines, which has proven to be effective in preventing *T. evansi* transmission to horses. Since horses often exhibit higher parasitemia and pathology compared to other animals, this approach could keep the horses healthy while protecting other animals as well, as they cannot act as reservoirs of parasites. Furthermore, to avert new *T. evansi* outbreaks akin to those in Spain and France, careful monitoring of animal import and export is essential. This involves implementing appropriate quarantine procedures and high PPV screening methods before relocation.

To date, a range of tests have been deployed to diagnose *T. evansi* infection worldwide. A meta-analysis of publications pertaining to *T. evansi* detection in domestic animals revealed that molecular tests were utilized in 6–28% of cases, while serological and parasitological tests were employed in 14–31% and 2–9% of cases, respectively [4]. Despite the numerous advantages offered by molecular diagnostic tests, various regions worldwide continue to rely on other tests. This is often due to limitations in field settings and the costs associated with molecular testing. However, as mentioned above, attention must be paid to the PPV of a diagnostic method to determine the most suitable diagnostic strategy. The ranking of diagnostic methods based on PPVs is as follows: molecular methods, serological methods, and parasitological methods. Among molecular methods, both LAMP- and RPA-based methods are gaining attention for their field applicability. While RPA holds certain advantages over LAMP as a rapid POC test, it still grapples with cost-related limitations. For practical implementation, the affordability of these tests is crucial for their wide-scale use in endemic countries. This necessitates the engagement of relevant production companies and funding entities to facilitate the commercialization of Surra diagnosis kits. The significance and efficacy of these diagnostic tests can be underscored through appropriate simulations that demonstrate how the containment of *T. evansi* spread can be achieved through efficient diagnosis and timely treatment. The development of multiplexed tests could also aid in species-specific treatment, although the additional cost must be carefully evaluated to ensure affordability.

Shifting focus, while considerable attention has been directed towards the diagnosis of Type A *T. evansi*, the development of a reliable POC test for Type B *T. evansi* is equally imperative. Although Type A *T. evansi* is predominant, the prevalence of Type B *T. evansi* might be greatly underestimated due to limited diagnostic efforts [80]. Also, the diagnosis of non-African atypical human trypanosomiasis caused by *T. evansi* could be underreported due to a lack of proper diagnostic tool availability, efforts, and campaigns. Therefore, expediting a faster and more efficient diagnosis of ‘all’ *T. evansi* infection types is necessary to prevent further transmission by enabling early-stage treatment. Indeed, in the context of a One Health approach, it is obvious that one cannot focus on a specific type of *T. evansi* infection, which is determined by the parasite type or host species. On the contrary, the control of *T. evansi* trypanosomosis needs to be prioritized as a fight against a near-global neglected disease in livestock that directly threatens the health of people living in close proximity to infected animals. Interestingly, in recent years, a huge success has been obtained in curbing the threat of human African trypanosomiasis (HAT), at least in the case of the most common *T. b. gambiense* variant of the disease, which historically has accounted for 98% of all HAT cases. The success in addressing *gambiense* HAT can be attributed to several factors: (i) the zoonotic nature of the disease, enabling effective control and surveillance of the human reservoir when coupled with vector control strategies, and (ii) a decade-long, robust collaboration between private and public funders, alongside well-coordinated South–North research interactions. This demonstrates that trypanosomiasis is a manageable disease. Unfortunately, the Surra situation has not garnered the same attention as gambiense HAT, primarily due to its challenging nature; its mechanical transmission, worldwide geographic distribution, and broader vector range make it a more formidable disease to tackle. Given its mode of transmission, *T. evansi* has the potential to evolve into a global problem, necessitating the development of novel control and surveillance measures [97]. Hopefully, with the development of high-PPV POC diagnostics currently being explored, this situation can be turned around in the near future.

## Figures and Tables

**Figure 1 microorganisms-12-00044-f001:**
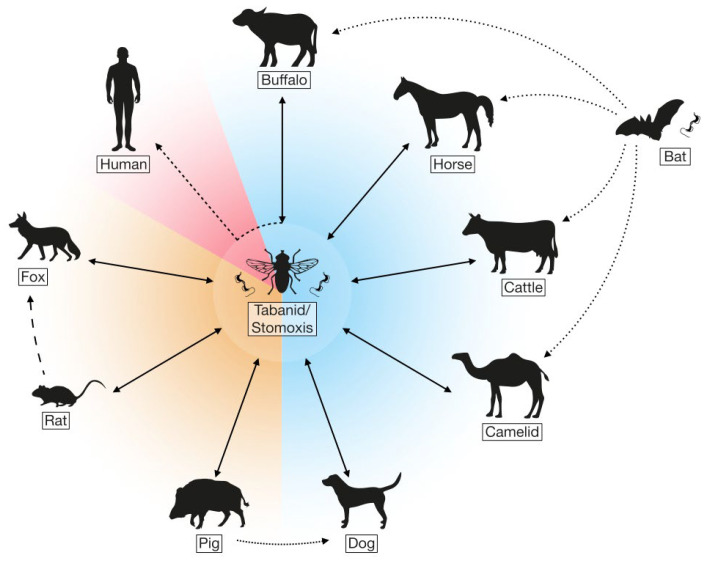
Mechanical transmission of *T. evansi* between a variety of mammals is mainly mediated by biting flies. The host species influenced by cattle vary based on geographical location and encompass livestock, companion animals, and wildlife. In South America, the vampire bat species *Desmodus rotundus* acts as both a reservoir host and aids in transmission. Human *T. evansi* infections, also referred to as aHT, have occasionally been reported in individuals who live in close proximity to infected livestock.

**Figure 2 microorganisms-12-00044-f002:**
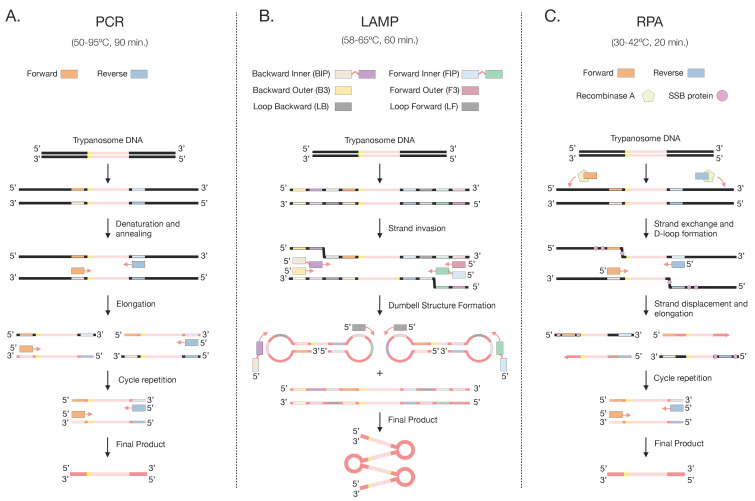
Overview of the main principles of the (**A**) PCR, (**B**) LAMP, and (**C**) RPA amplification methods. Squares matching the same colors as the primers but with a lighter shade correspond to the complementary sequences of these.

**Figure 3 microorganisms-12-00044-f003:**
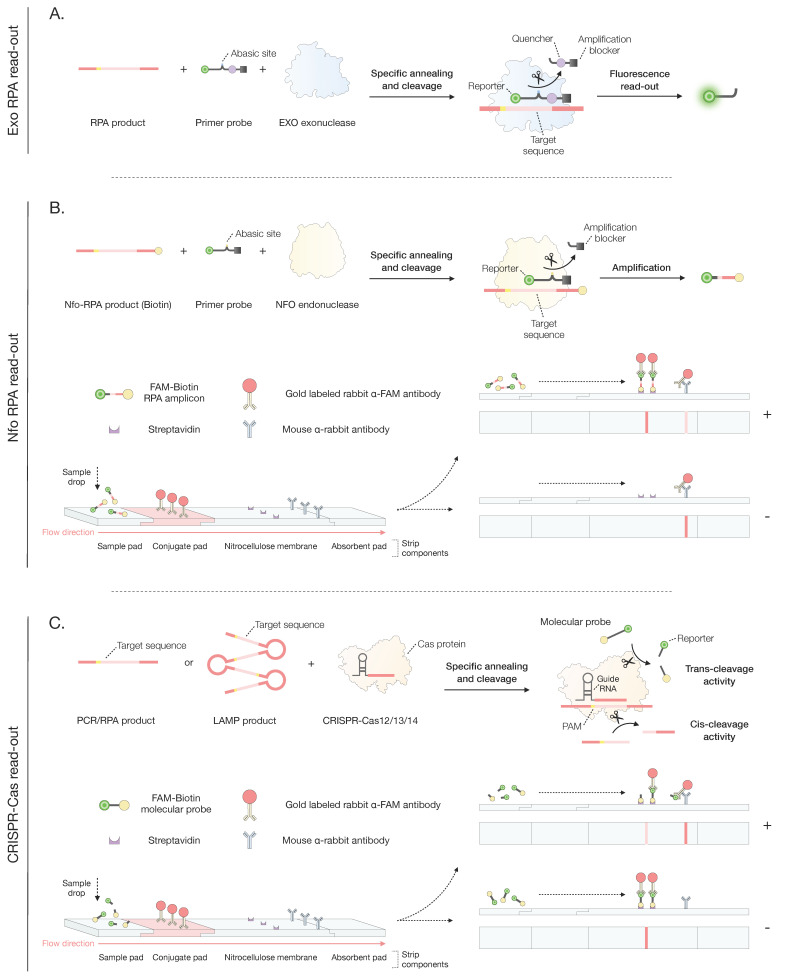
Overview of different readout methods for PCR, LAMP, and RPA. (**A**) The exoRPA readout is based on the detection of the fluorescence released after the cleavage of a FAM quencher–blocker probe by exonuclease III. The EXO protein cleaves the probe at the abasic site contained between the FAM and the quencher only after the specific binding of the probe to the target DNA. (**B**) The Nfo RPA readout is based on the specific detection of the FAM-Biotin amplicon by gold-labeled antibodies contained on a lateral flow strip. For this, a standard RPA reaction is first performed, with the only exception of the reverse primer being linked to a biotin molecule. After amplification, a probe consisting of an FAM reporter and an amplification blocker is annealed specifically to the target RPA amplicon. Then, the Nfo endonuclease cleaves the probe at the abasic site, enabling the occurrence of a second amplification using the cleaved FAM probe as a primer. Finally, the newly synthesized FAM-biotin amplicon is poured into a lateral flow strip, where it will be detected and visualized by specific antibodies. (**C**) The CRISPR-Cas readout is based on the detection of cleaved FAM reporters after the specific detection of PCR, LAMP, or RPA amplicons by the CRISPR-Cas machinery (Figure 3). Shortly, the Cas endonuclease, together with a sgRNA, detects a specific region contained in the amplicons, resulting in their cis-cleavage into two fragments. As a result, the activated Cas protein undertakes a non-specific cleavage of surrounding off-target single-stranded DNA/RNA probes (containing an FAM and a biotin molecule), so-called trans-cleavage activity. Finally, the cleaved FAM probes are poured into a lateral flow strip, where they will be detected and visualized by specific antibodies.

## Data Availability

Not applicable.

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
