# Peer review of "Recent Progress in the Detection of Surra, a Neglected Disease Caused by Trypanosoma evansi with a One Health Impact in Large Parts of the Tropic and Sub-Tropic World"

_microorganisms, 2023, doi:10.3390/microorganisms12010044_

Round 1
Reviewer 1 Report
Comments and Suggestions for Authors
Report on Kim et al (Microorganisms 2023)
This is a timely, interesting, knowledgeable and well-written review on the diagnosis of surra. It takes a practical approach to the applicability of the different diagnostic options, considering always the limitations, costs, equipment needed and even time-to-result of each test. It is also for the most part quite easy to read.
I have very few comments to make apart form a few suggestions that the authors could consider.
Line 55: ‘accidental importation’. Perhaps change for ‘unwitting’ or another synonym? Importation is not accidental.
Line 67: I suggest insect faeces rather than stool, which Prof Magez will know refers to a type of primitive toilet that the insects probably don’t use.
Line 83-84. ‘Their mouthpart are designed to facilitate the transmission of T. evansi’. I don’t think ‘designed’ is an appropriate word here. I could suggest ‘optimal’.
Last paragraph of section2. I would have liked to see more detail on the zoonotic potential of T. evansi infection in the One Health context. There was a link with a genetic defect, specifically the lack of Apoliprotein L1 (PMID: 17192540; DOI: 10.1056/NEJMoa063265). I also think this paper ought to mentioned: PMID: 24069464 ;DOI: 10.1371/journal.pntd.0002256.
Line 128: I would use ‘diminazene aceturate’ as Berenil is only one trade name this product is sold under. I also think that it should be mentioned that diminazene is also blood-stage only, with no effect on CNS infections. Other treatments used in surra and dourine are melarsomine/cymelarsan and quinapyramine.
Last section of sentence of section 4.1. I think it should be spelled out why correct identification of trypanosome species is so important, namely that treatment must be tailored to the species, which unequal inherent susceptibility to the different drugs. See for instance this recent paper. doi: 10.3390/ijms23052844; PMID: 35269985. Because of that I would like a bit more mention of the possibilities for multiplexing the various diagnostic tools. For instance in South America, there is a large overlap in areas infected with T. evansi and T. vivax. Testing for only one or the other is not good enough. Equally, I suspect that T. evansi is severely underreported in Africa, not being recognised as such. In short, please address the diagnostic challenges of T. evansi diagnosis in areas/regions where multiple trypanosome species are endemic.
Line 242. Yes, real-time PCR will quantify the infection but it might be mentioned here that the parasitaemia is highly variable and that the level on any particularly day does not provide a clinically determinative number.
I would suggest a Figure, with the details of PCR, LAMP and RPA (Exo and LF) side-by-side in a diagram format.
Line 441: typo, delete ‘ref’
Line 443, in the summing up the authors state that the recent success in tackling HAT shows that trypanosomiasis can be controlled. This seems rather simplistic although the optimism is welcome. However, whereas HAT transmission can be much-reduced by tsetse control, such insect control is much harder for surra, its transmission being enabled by multiple common species of biting flies. Also, the control of HAT is essentially of the gambiense variety only, which has a negligible animal reservoir, whereas T. evansi does have a non-domestic animal reservoir. I think it is important to point out some of these challenges, as well as the tremendous effort on HAT case finding that was necessary for the near eradication of gambiense HAT.
Author Response
Hereby we would like to resubmit the paper entitled "Recent progress in the detection of Surra, a neglected disease caused by Trypanosoma evansi with a One Health impact in large parts of the tropic and sub-tropic world". We have answered all queries by the reviewers as outlined below and included 2 comprehensive figures to increase the attractiveness of the paper.
All answers are outlined below and have been highlighted in the text as well. Please refer to the line numbers mentioned in the answers, mentioning where changes were made in the text.
Reviewer 1
- Line 55: ‘accidental importation’. Perhaps change for ‘unwitting’ or another synonym? Importation is not accidental.
Line 58. Word replaced for clarity.
- Line 67: I suggest insect faeces rather than stool, which Prof Magez will know refers to a type of primitive toilet that the insects probably don’t use.
Line 79. Word replaced for clarity.
- Line 83-84. ‘Their mouthpart are designed to facilitate the transmission of T. evansi’. I don’t think ‘designed’ is an appropriate word here. I could suggest ‘optimal’.
Line 90. Word replaced for clarity.
- Last paragraph of section2. I would have liked to see more detail on the zoonotic potential of T. evansi infection in the One Health context. There was a link with a genetic defect, specifically the lack of Apoliprotein L1 (PMID: 17192540; DOI: 10.1056/NEJMoa063265). I also think this paper ought to mentioned: PMID: 24069464 ;DOI: 10.1371/journal.pntd.0002256.
Line 107-114. More information about these atypical infections was added.
- Line 128: I would use ‘diminazene aceturate’ as Berenil is only one trade name this product is sold under. I also think that it should be mentioned that diminazene is also blood-stage only, with no effect on CNS infections. Other treatments used in surra and dourine are melarsomine/cymelarsan and quinapyramine.
Line 141-143. Addition of more Surra treatments, full name of drugs and specification of its blood-stage use.
- Last section of sentence of section 4.1. I think it should be spelled out why correct identification of trypanosome species is so important, namely that treatment must be tailored to the species, which unequal inherent susceptibility to the different drugs. See for instance this recent paper. doi: 10.3390/ijms23052844; PMID: 35269985. Because of that I would like a bit more mention of the possibilities for multiplexing the various diagnostic tools. For instance in South America, there is a large overlap in areas infected with T. evansi and T. vivax. Testing for only one or the other is not good enough. Equally, I suspect that T. evansi is severely underreported in Africa, not being recognised as such. In short, please address the diagnostic challenges of T. evansi diagnosis in areas/regions where multiple trypanosome species are endemic.
Lines 190-196. More information about the matter was added.
- Line 242. Yes, real-time PCR will quantify the infection but it might be mentioned here that the parasitaemia is highly variable and that the level on any particularly day does not provide a clinically determinative number.
Line 265-267. Modified and added for clarity.
- I would suggest a Figure, with the details of PCR, LAMP and RPA (Exo and LF) side-by-side in a diagram format.
Section 5.1 and 5.2.1. A figure with the details of standard PCR, LAMP and RPA was added. Also, a second figure was included where displayed the different read-out methods for RPA (Exo and Nfo-LF) as well as the CRISPR-Cas read-out for PCR, LAMP and RPA.
- Line 441: typo, delete ‘ref’
Line 495. Deleted.
- Line 443, in the summing up the authors state that the recent success in tackling HAT shows that trypanosomiasis can be controlled. This seems rather simplistic although the optimism is welcome. However, whereas HAT transmission can be much-reduced by tsetse control, such insect control is much harder for surra, its transmission being enabled by multiple common species of biting flies. Also, the control of HAT is essentially of the gambiense variety only, which has a negligible animal reservoir, whereas T. evansi does have a non-domestic animal reservoir. I think it is important to point out some of these challenges, as well as the tremendous effort on HAT case finding that was necessary for the near eradication of gambiense HAT.
Line 497-507. Agreed and modified as following: Interestingly, in recent years, a huge success has been obtained in curbing the threat of Human African Trypanosomiasis (HAT), at least in case of the most common T. b. gambiense variant of the disease, that historically has accounted for 98% of all HAT cases. The success in addressing gambiense HAT can be attributed to several factors: (i) the zoonotic nature of the disease, enabling effective control and surveillance of the human reservoir when coupled with vector control strategies, and (ii) a decade-long, robust collaboration between private and public funders, alongside well-coordinated South-North research interactions. This demonstrates that trypanosomiasis is a manageable disease. Unfortunately, the Surra situation has not garnered the same attention as gambiense HAT, primarily due to its challenging nature-mechanical transmission, worldwide geographic distribution, and a broader vector range make it a more formidable disease to tackle. Given its mode of transmission, T. evansi has the potential to evolve into a global problem, necessitating the development of novel control and surveillance measures.
Reviewer 2 Report
Comments and Suggestions for Authors
The manuscript microorganisms-2735935 entitled of " Recent progress in the detection of Surra, a neglected disease caused by Trypanosoma evansi with a One Health impact in large parts of the tropic and sub-tropic world. " by Jeongmin et al provides the recent progress on detecting methods for Trypanosoma evansi. Although of interest, I feel that the manuscript requires Major revision before it is suitable for publication.
1. This review should focus on the recent progress of detection methods for Trypanosoma evansi, section of “2. Transmission and Pathology and 3. Control and Treatment of Surra can be briefly described in section of 1. Introduction
2. Recent progress should be compared with other Trypanosoma spp.detection methods.
Author Response
Hereby we would like to resubmit the paper entitled "Recent progress in the detection of Surra, a neglected disease caused by Trypanosoma evansi with a One Health impact in large parts of the tropic and sub-tropic world". We have answered all queries by the reviewers as outlined below and included 2 comprehensive figures to increase the attractiveness of the paper.
All answers are outlined below and have been highlighted in the text as well. Please refer to the line numbers mentioned in the answers, mentioning where changes were made in the text.
Reviewer 2
- This review should focus on the recent progress of detection methods for Trypanosoma evansi, section of “2. Transmission and Pathology and 3. Control and Treatment of Surra can be briefly described in section of 1. Introduction
In-depth knowledge of the progress of detection methods for Trypanosoma evansi has been added. Please refer to figure 2, 3 and tables 1 and 2.
- Recent progress should be compared with other Trypanosoma spp.detection methods.
Line 395. Molecular diagnostic tests table was modified, including other Trypanosoma spp. methods.
Line 238. A new table including the serological tests for the diagnosis of T. evansi was also included.
Reviewer 3 Report
Comments and Suggestions for Authors
Few errors were detected and are highlighted in the attached revised manuscript.
Overall the manuscript is important as it focuses on Surra one of neglected tropical diseases in world. I'm however missing the One Health as the authors do not explain how T. evansi can be detected in the various vectors they only focused on how it was detected in mammalian hosts.

The encouraged to avoid the use of acronyms and abbreviations to start sentences.
Author Response
Hereby we would like to resubmit the paper entitled "Recent progress in the detection of Surra, a neglected disease caused by Trypanosoma evansi with a One Health impact in large parts of the tropic and sub-tropic world". We have answered all queries by the reviewers as outlined below and included 2 comprehensive figures to increase the attractiveness of the paper.
All answers are outlined below and have been highlighted in the text as well. Please refer to the line numbers mentioned in the answers, mentioning where changes were made in the text.
Reviewer 3
- Write the genus in full as it starts the sentence
Line 17. Genus written in full at the start of the sentence.
- rewrite to "domestic and wild"
Line 36. Word order changed.
- Write in full
Line 41. Genus written in full at the start of the sentence.
- same as above comment
Line 73. Genus written in full at the start of the sentence.
- rewrite to "Mechanical transmission of T. evansi between a variety of mammals is …."
Line 81. Rewritten.
- confusing to read, please rephrase
Line 81. Rephrased.
- rewrite to " acts as both the reservoir host as well as a transmission vector”
Line 81. Rewritten.
- of Africa
Line 179. Modified.
- rewrite to " The RoTat 1.2-based diagnostic tests detect antibodies in the serum that indicate a Type A T. evansi infection and they include tests such as the Card Agglutination Test for T. evansi (CATT/T. evansi), the Latex Agglutination Test for T. evansi (LATEX/T. evansi), and the Enzyme-Linked Immunosorbent Assay for T. evansi (ELISA/T. evansi).
Line 206-210. Rewritten.